# A VEL3 histone deacetylase complex establishes a maternal epigenetic state controlling progeny seed dormancy

Xiaochao Chen[1], Dana R. MacGregor [1,2], Francesca L. Stefanato [1], Naichao Zhang[1,3], Thiago Barros-Galvão [1] & Steven Penfield [1] ✉

Mother plants play an important role in the control of dormancy and dispersal characters of their progeny. In Arabidopsis seed dormancy is imposed by the embryo-surrounding tissues of the endosperm and seed coat. Here we show that VERNALIZATION5/VIN3-LIKE 3 (VEL3) maintains maternal control over progeny seed dormancy by establishing an epigenetic state in the central cell that primes the depth of primary seed dormancy later established during seed maturation. VEL3 colocalises with MSI1 in the nucleolus and associates with a histone deacetylase complex. Furthermore, VEL3 preferentially associates with pericentromeric chromatin and is required for deacetylation and H3K27me3 deposition established in the central cell. The epigenetic state established by maternal VEL3 is retained in mature seeds, and controls seed dormancy in part through repression of programmed cell death-associated gene *ORE1*. Our data demonstrates a mechanism by which maternal control of progeny seed physiology persists post-shedding, maintaining parental control of seed behaviour.

In most angiosperms double fertilization creates a diploid embryo and triploid endosperm tissues respectively. During the control of *Arabidopsis thaliana* seed development mother and father generate unique epigenetic landscapes in endosperm that lead to parent-of-origin gene expression via genomic imprinting[1,2]. In the central cell of female gametophyte PRC2 establishes H3K27me3-mediated silencing at maternal alleles to produce imprinted paternally expressed genes (PEGs)[3]. Loss of maternal PRC2 activity leads to overexpression of the maternal alleles of PEGs and failure of endosperm cellularisation[4,5]. These epigenetic landscapes are important in resource conflicts between mother, father and offspring[6,7], because the timing of endosperm cellularisation affects seed size and therefore resource availability.

Seed dormancy is also controlled by the endosperm and is subject to parental conflicts because of the role of dormancy in risky maternal dispersal strategies. Coat-imposed seed dormancy requires the establishment of barriers to oxygen uptake into the seed and the action of the plant hormone abscisic acid, which is transported from the endosperm to the embryo to prevent germination[8,9]. Coat-imposed seed dormancy additionally requires the silencing of paternal alleles, via non-canonical RNA-directed DNA methylation targeted to transposons in the male germline, particularly at the *ALLANTOINASE* locus[10]. Although imprinted genes can be expressed at any stage of seed development and germination, the timing of chromatin remodeling events that control dormancy and germination is less clear. The H3K27me3 demethylase REF6 is required for gene activation during germination[11] and imprinting can also be lost during dormancy loss in the mature seed[12]. However, some genetic evidence suggests that some chromatin modellers act earlier in endosperm development, or even in the central cell to affect seed dormancy[11]. Here, we show that mothers maintain progeny dormancy via inheritance of dormancy-inducing epi-alleles in the endosperm.

[1]Department of Crop Genetics, John Innes Centre, Norwich Research Park, Norwich NR4 7UH, UK. [2]Protecting Crops and the Environment, Rothamsted Research, Harpenden AL5 2JQ, UK. [3]Henan University, Jinming Road, Kaifeng, Henan, China. ✉e-mail: steven.penfield@jic.ac.uk

## Results

### Genome-wide association links the *VEL3* locus to seed dormancy control

To understand variation in the environmental control of seed dormancy, we set seeds of 271 accessions at two different temperatures and sowed them without or with stratification at 4 °C or 16 °C prior to germination (Fig. 1a and Supplementary Data 1). We used Genome-Wide Association to identify dormancy-associated genes[13], many of

which coincided with previously identified *DELAY OF GERMINATION* (*DOG*) loci[14,15]. Conditions which caused variation among the most dormant accessions revealed the *DOG1* and *DOG6* loci, whereas conditions that caused higher germination identified a locus close to the *PHYTOCHROME B* (*phyB*) in addition to other candidate genes (Fig. 1b). We scored seed dormancy in several T-DNA mutants disrupting genes associated with dormancy in our GWAS. Two alleles in *ANAC60* showed enhanced dormancy when set at 22 °C (Supplementary Fig. 1), as

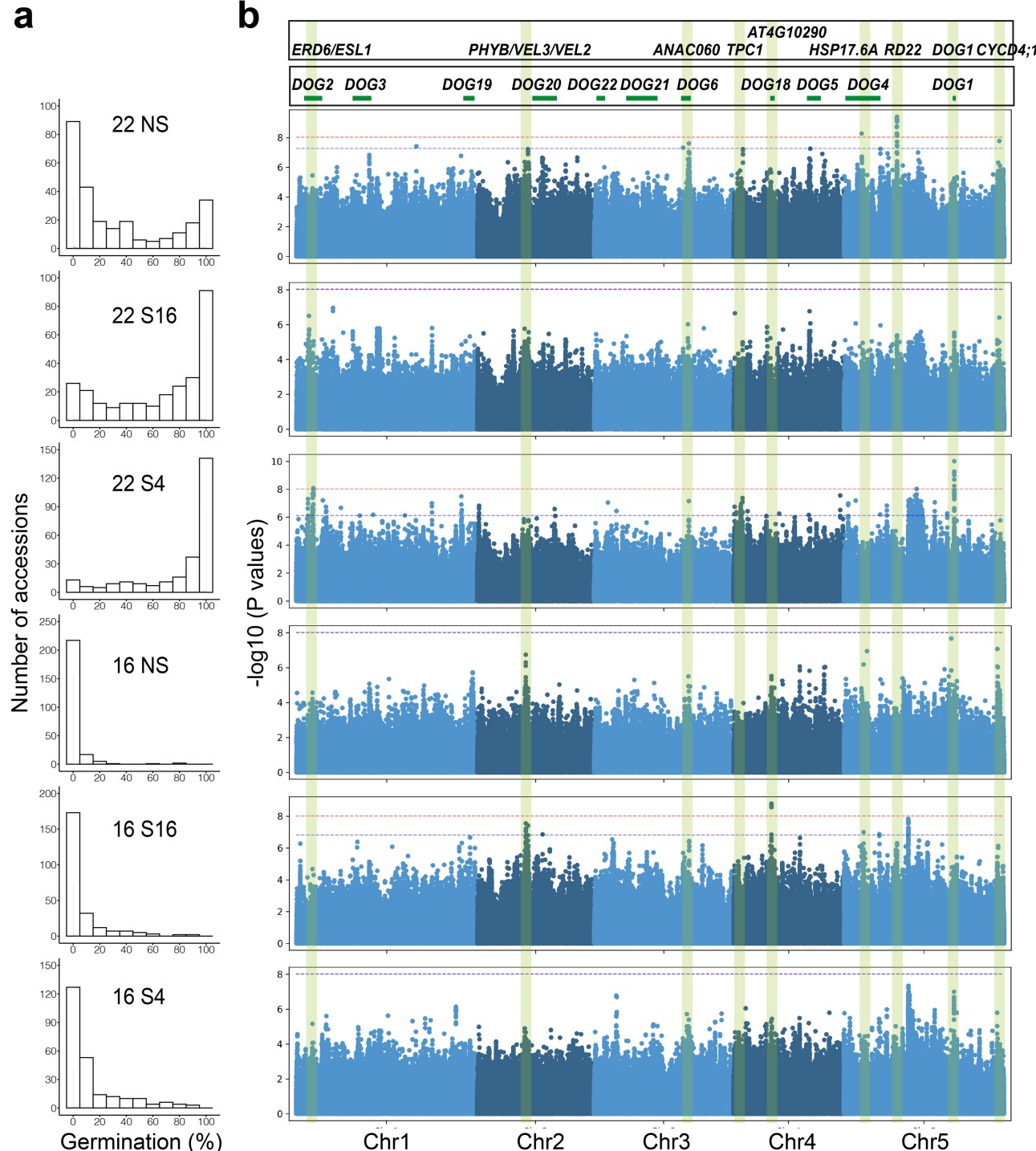

**Fig. 1 | Genome-wide association identifies a locus on chromosome 2 with a role in seed dormancy control. a** Percentage of accessions with different germination frequencies when grown at either 16 °C or 22 °C and sown either without prior stratification (NS) or after dark stratification at 4 °C (S4) or at 16 °C (S16). **b** Manhattan plots to show the distribution of dormancy-associated SNPs across treatments, relative to previously identified *DOG* loci derived by Kruskal–Wallis method.

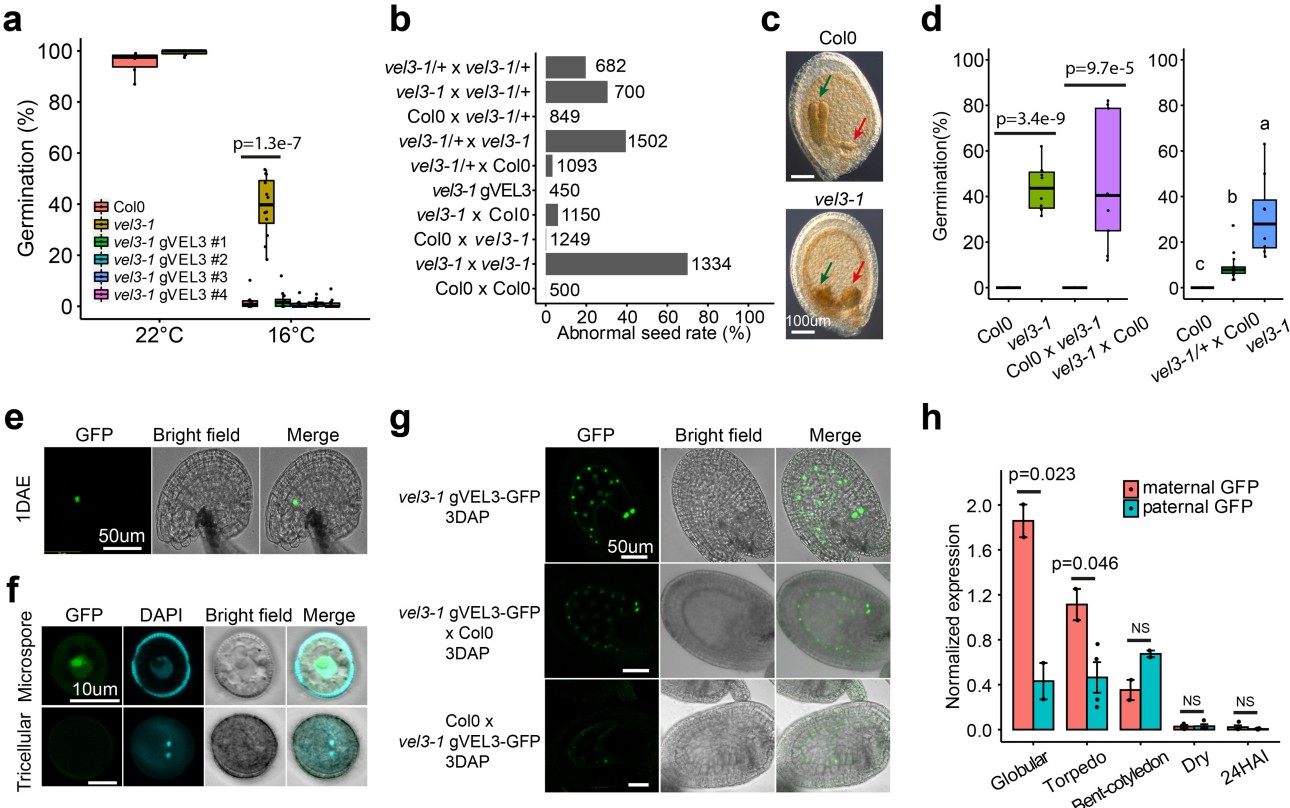

**Fig. 2 | The role of VEL3 in the control of endosperm development and maternal control of primary seed dormancy. a** VEL3 is important for seed dormancy establishment when seeds are set at low temperatures. *vel3-1* seeds show a clear low-dormancy phenotype, which is complemented by transformation with the *VEL3* gene sequence in 4 independent transgenic lines *vel3-1 gVEL3*. *p*-value is derived from the two-sided *t*-test. Boxplots indicate minimum and maximum values, as well as 25th, 50th and 75th percentiles. **b** Mutant *vel3-1* seeds exhibit high-frequency seed abortion. Reciprocal crosses (always maternal x paternal in this paper) show that both the maternally derived and paternally derived VEL3 gene copies can act to promote normal seed development, and that complemented *vel3-1* mutants (*vel3-1 gVEL3*) show normal seed development. Total number of seeds analyzed for each genotype are shown. **c** Representative DIC images of WT and *vel3-1* seeds, demonstrating retardation of embryo development in *vel3-1* (blue arrow) and an enlarged chalazal endosperm (red arrow). The experiments were repeated ten times. **d** Germination of freshly harvested seeds show that maternal VEL3 controls seed dormancy from the female gametophyte. Loss of maternal VEL3 is sufficient to affect seed dormancy whereas heterozygous mothers pollinated with WT pollen show an intermediate phenotype. Data represents minimum and maximum values as well we 25th, 50th and 75th quartiles of 9 to 10 biological replicates per genotype. *p*-values for significant differences using 2-tailed *t*-test are shown. **e** Confocal microscopy and corresponding brightfield image showing *vel3 gVEL3-GFP* in the central cell of the unfertilized emasculated ovule 1 day after anthesis. DAE: Days after emasculation. Scale bar 50 μm. The experiments were repeated three times. **f** Confocal microscopy and corresponding brightfield image showing transient *vel3 gVEL3-GFP* in the microspore, which was absent from mature pollen. Scale bars 10 μm. The experiments were repeated three times. **g** Confocal images of *vel3 gVEL3-GFP* seeds showing VEL3-GFP in the nuclei of the developing endosperm from homozygous plants, or in seeds derived from reciprocal crosses to show the activity of the maternally derived and paternally derived gene copies. Scale bars 50 μm. The experiments were repeated five times. **h** Expression of the paternally derived and maternally derived VEL3 gene copies during seed development, using F1 seeds generated by reciprocal crosses between WT and *vel3-1 vel3 gVEL3-GFP* by qRT-PCR. Mean and standard error of two to four biological replicates are shown (see source data), with *p*-values derived from 2-tailed *t*-tests. Seeds were standardized by embryo development stage, globular (glob), torpedo (torp), bent cotyledon (bent) mature dry seeds and 24 h after imbibition (24HAI).

recently described[16]. In the *phyB* region the most strongly dormancy-associated SNPs were approximately 20 kb distal to the *phyB* gene, in a locus with two seed-expressed homologs of *VIN3, VEL2* and *VEL3* (Supplementary Fig. 2a), and we found that loss of VEL3 function leads to low dormancy (Supplementary Fig. 1; Fig. 2a; we could not isolate *vel2* T-DNA mutants to analyze). There is weak linkage disequilibrium between *phyB* and *VEL2/3* both of which were characterized by 3 major haplotypes consistent with those previously described at the *phyB* locus (Supplementary Fig. 2b and ref. 17). ANOVA indicated that the high germination accessions most frequently contained a single haplotype of *phyB* previously described to be a weak allele controlling red light responses in seedlings (Supplementary Fig. 2c, d and Supplementary Data 2). Reciprocal crosses revealed a strong maternal bias to *phyB* action in seed dormancy control, with loss of the paternal allele having a limited impact on seed germination (Supplementary Fig. 2e), although we cannot rule out an additional role for variation at VEL3 in seed dormancy control.

## The role of VEL3 in the maternal control of primary seed dormancy

The low-dormancy phenotype in *vel3-1* was found in a second allele, *vel3-2*, in the WS background (Supplementary Fig. 3a). We also observed a high frequency of aborted seeds in *vel3* mutants and did not observe endosperm cellularisation in *vel3* mutants. (Fig. 2b, c and Supplementary Fig. 3a–e). To confirm that disruption of *VEL3* was responsible for these phenotypes we complimented *vel3-1* with the TAIR10-annotated *VEL3* cDNA (Supplementary Fig. 4a, b). This did reduce seed abortion in some lines but could not completely rescue the phenotype (Supplementary Fig. 4c, d). Amplification and sequencing of the *VEL3* cDNA instead revealed a intron-exon structure, previously annotated in Ler[18] that when introduced into *vel3-1* (named *vel3-1 gVEL3*) completely rescued both the dormancy and seed abortion phenotypes (Fig. 2a, b and Supplementary Fig. 4a, b, e, f). Reciprocal crosses indicated a strong maternal bias to the control of seed dormancy by VEL3 in freshly harvested seed (Fig. 2d). To test whether

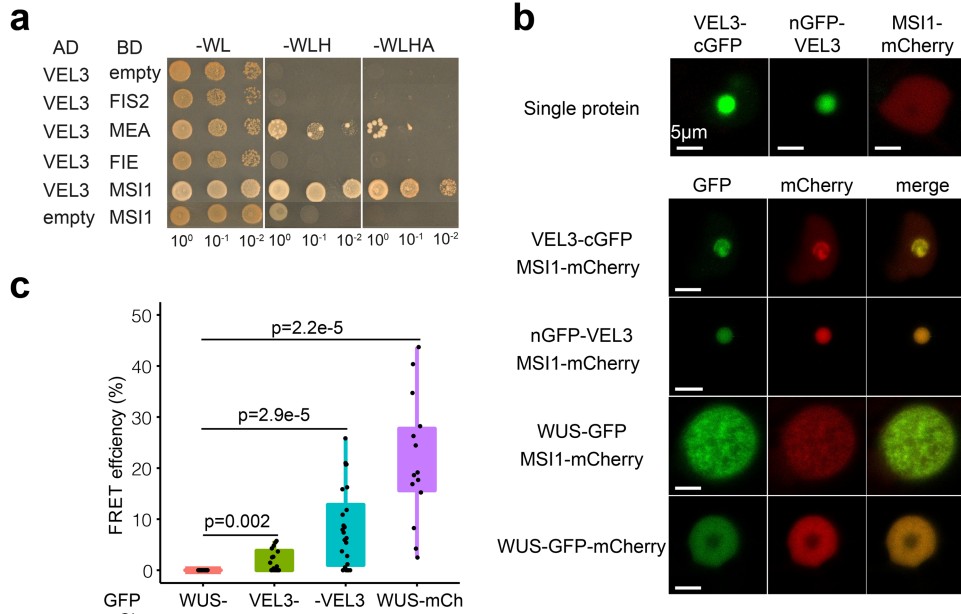

**Fig. 3 | VEL3 interacts with MSI1 and can direct MSI1 to the nucleolus. a** Yeast 2-hybrid assays show a robust interaction with the PRC2 component MSI1. MEA shows autoactivation in our assays (Supplementary Fig. 3e). **b** VEL3 expression affects the sub-nuclear localization of MSI1. Nuclear images of *Nicotiana* cells expressing VEL3 fused at either the N- or C-terminus to GFP with cells expressing either VEL3 alone, MSI alone or VEL3 and MSI1. Co-expression of MSI1 with the control nuclear protein WUSCHEL (WUS) is shown for comparison. VEL3-: VEL3-

cGFP; -VEL3:nGFP-VEL3. Scale bars 5 μm. The experiments were repeated twice. **c** FRET shows a close interaction between VEL3 and MSI. Comparison with the positive control using the heterodimerising WUS protein, and negative control combination MSI1 and WUS, data represents minimum and maximum values, as well as 25th, 50th, and 75th quartiles. Sample sizes are included in source data ($N = 16,16,24,14$ from left to right). *p*-values are derived from the two-sided *t*-test.

the *VEL3* gene acts in the maternal sporophyte or female gametophyte we pollinated heterozygous *vel3-1* /+ plants with wild-type pollen. These showed an intermediate phenotype consistent with a gametophytic origin of VEL3 activity (Fig. 2d). In contrast, the seed abortion phenotype could be rescued by either a functional maternal or paternal *VEL3* gene, showing that VEL3 functions in the zygote to control this trait (Fig. 2b and Supplementary Fig. 3c).

We also complemented the *vel3-1* mutant with the *VEL3* cDNA under the control of its own promoter fused to GFP (*vel3 gVEL3-GFP*; Supplementary Fig. 4b, e). We first observed nuclear VEL3-GFP in the central cell of the ovule (Fig. 2e) and transiently during microgametogenesis, but not in mature pollen. VEL3-GFP was primarily located in the nucleolus but a lower GFP signal was observed in the nucleoplasm (Fig. 2f). During seed development VEL3-GFP was found exclusively in the coenocytic endosperm, with expression declining prior to cellularisation (Fig. 2g). We observed stronger VEL3-GFP expression from the maternal allele, either using confocal microscopy or qRT-PCR in $F_1$ seeds derived from reciprocal crosses (Fig. 2g, h). Therefore, we concluded that VEL3 controls seed development and seed dormancy by acting in central cell.

## VEL3 co-localizes with MSI1 in the nucleolus
VEL3 is a member of the VIN3 subclade of PHD finger proteins that act as PRC2 accessory proteins during the vernalization response[19,20] although VEL3 itself lacks a PHD domain. Furthermore, disruption of maternal PRC2 activity in the endosperm results in seed abortion due to mis-expression of paternally expressed genes (PEGs) from the maternal genome including *PHERES1*[5]. The *vel3* seed abortion syndrome follows a similar path to loss of PRC2, with delayed seed development and enlarged chalazal endosperm (Fig. 2c and Supplementary Fig. 3d). Therefore, we tested whether VEL3 could directly bind seed-specific FIS-PRC2 components using Yeast two-hybrid (Y2H). Using Y2H we found a putative interaction with MULTICOPY SUPPRESSOR OF IRA 1 (MSI), but not with other FIS-PRC2 subunits

(Fig. 3a and Supplementary Fig. 3f). To confirm this interaction *in planta* we co-expressed VEL3-GFP and MSI1-mCherry in *Nicotiana benthamiana* (Fig. 3b). As in Arabidopsis we observed VEL3-GFP primarily in the nucleolus, while MSI1 was localized primarily to the nucleoplasm. Co-infiltration of VEL3 with MSI1 resulted in MSI1 co-localizing in the nucleolus with VEL3, suggesting that the VEL3 protein can affect the sub-nuclear localization of MSI1. Using FRET we confirmed that VEL3-GFP interacts with MSI1 *in planta* (Fig. 3c). The interaction could also be confirmed by IP-MS (see Supplementary Data 4).

## VEL3 is required for pericentromeric H3K27me3 deposition
To further test whether VEL3 has a role in maternal PRC2 activity in the endosperm we used ChIP-seq to test whether VEL3 is required for H3K27me3 deposition in the endosperm. ChIP-seq in pure endosperm samples is possible in mature seeds because of the death of seed coat cells during maturation[10]. Firstly, we found H3K27me3 deposition in the Col0 endosperm takes place during early endosperm development because the H3K27me3 distribution in the mature endosperm closely resembles that described previously at the globular stage (Fig. 4a)[21]. However, in the *vel3-1* endosperm H3K27me3 was globally reduced compared to WT (Fig. 4a, b). H3K27me3 is deposited in the central cell at demethylated sites affected by the activity of the DNA glycosylase DEMETER (DME)[3] and is concentrated at pericentromeric regions. Using ChIP-seq to characterize the binding sites genome-wide of VEL3-GFP in globular stage seeds we found that VEL3 also preferentially associates with transposon-rich areas at pericentromeric regions, with a substantial overlap between VEL3 binding and the occurrence of H3K27me3 (Fig. 4c). Thus, VEL3 is required for the establishment or maintenance of H3K27me3 at pericentromeric regions in the endosperm. The similar genome-wide distribution of H3K27me3 in the globular and mature endosperm also supports the conclusion that H3K27me3 established in early endosperm development is rarely re-

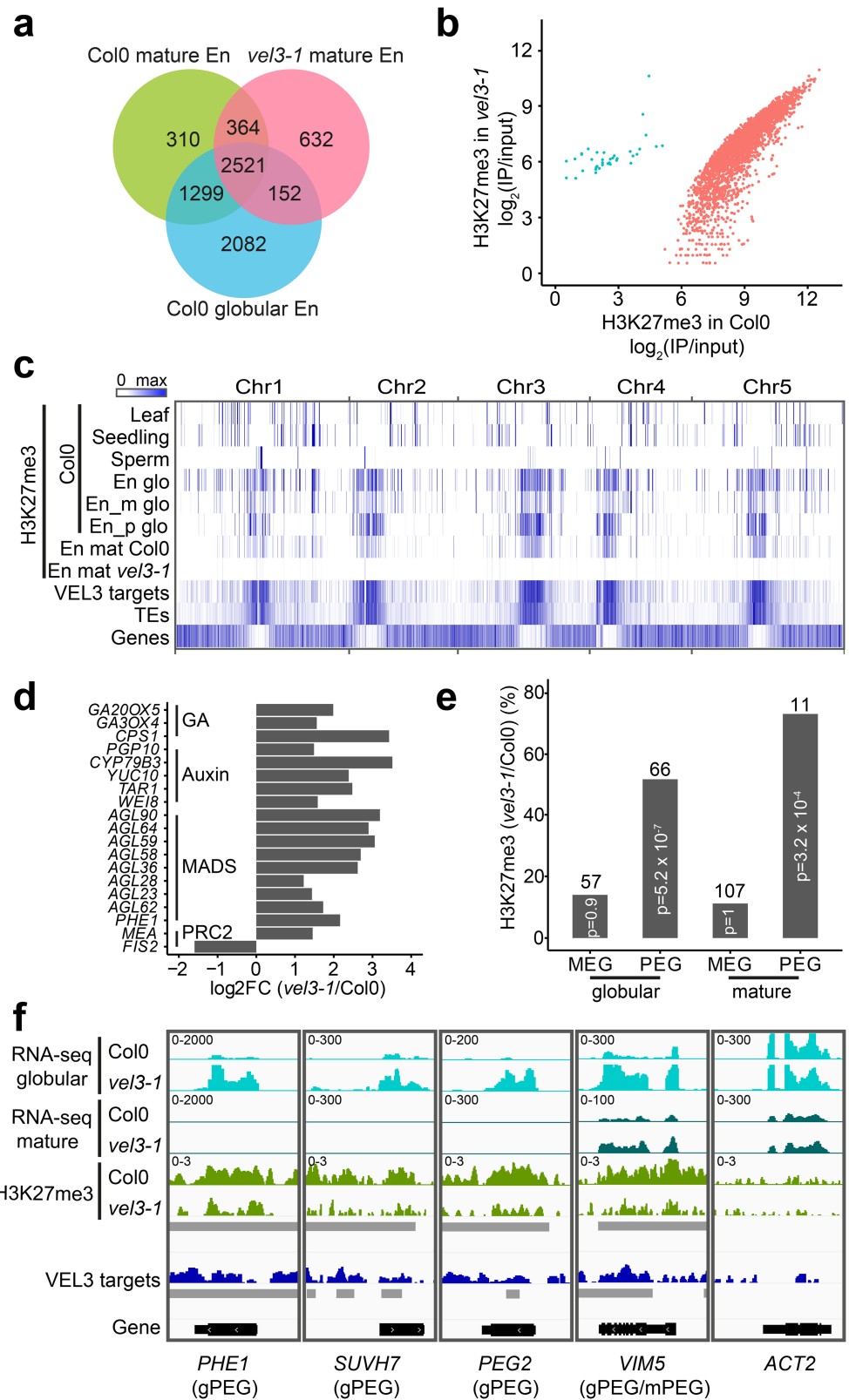

programmed later in seed maturation (Fig. 4c). Pericentromeric heterochromatin formed in the central cell is characterized by the overlapping presence of H3K27me3 and H3K9me2[22]. We found that VEL3 is required for H3K27me3 regardless of whether H3K9me2 is also present at a locus, and that there was no obvious relationship between VEL3 and H3K9me2 deposition (Supplementary Fig. S5). A subset of these can be targeted by REF6, which can also affect seed

dormancy (Supplementary Fig. S5) although it is not clear whether REF6 acts before or after VEL3 to affect seed dormancy.

We also used RNA-seq to compare gene expression between WT and *vel3-1* seeds at globular stage 7 days after anthesis at 16 °C. We found that a suite of previously described PEGs are upregulated in *vel3-1*, including *PHE1*, and several genes known to be mis-expressed in seeds lacking FIS-PRC2 including MADS box genes, auxin biosynthetic

**Fig. 4 | VEL3 is required for the establishment of a persistent pattern of H3K27me3 deposition in the endosperm. a** Chromatin immunoprecipitation to compare the number of genes marked with H3K27me3 in mature Col-0 and *vel3-1* endosperms, compared to globular stage endosperm tissues[21], showing conservation of marks across endosperm development and dependency on VEL3. **b** Quantitative effect of VEL3 loss on H3K27me3 at genes in the mature endosperm shows a reduction in H3K27me3 genome wide at gene loci. Blue shows loci with increased H3K27me3 in *vel3-1* vs. Col-0, red shows loci with reduced H3K27me3 in *vel3-1*. **c** Genome-wide comparison of VEL3-GFP chromatin distribution with H3K27me3, location of genes and transposons (TEs), demonstrating the general requirement for VEL3 for the targeting of H3K27me3 to pericentromeric regions bound by the VEL3 protein. En glo: globular endosperm; En_m glo: maternally derived endosperm alleles at globular stage; En_p paternally derived alleles; TE: transposable element; En mat: mature endosperm. **d** RNA-seq analysis of WT vs. *vel3-1* mutant seeds at the globular stage at 16 °C. Expression of genes known to be mis-regulated in endosperms lacking FIS2-PRC2 are shown, demonstrating a role for VEL3 in regulating genes silenced by PRC2. All genes shown are significantly mis-expressed in *vel3-1* compared to WT (FDR < 0.05) in three biological replicate experiments. **e** VEL3 is required for H3K27me3 deposition and the establishment of PEGs in the endosperm. Chart shows the percentage of PEGs and MEGs at globular stage and mature stage endosperms that show VEL3-dependent H3K27me3 deposition. The number of genes in each category is shown above the bars. Significantly enriched loci were determined using two-sided Fisher's exact test. **f** VEL3 is required for normal expression levels of PEGs. RNA-seq comparing WT and *vel3-1* mutants shown with H3K27me3 levels and VEL3 protein binding sites at four previously characterized PEGs and the control gene *ACTIN2* (*ACT2*). gPEGs are PEGs at the globular stage, mPEGs are PEGs in mature seeds. Gray bars indicate significant regions (*p* < 0.05) identified by DiffBind (for H3K27me3 between Col0 and *vel3-1*) or MACS2 (for VEL3 ChIP between IP and input).

genes and the auxin transporter *PGP10* (Fig. 4d and Supplementary Data 3). PRC2 activity is critical for silencing maternal alleles leading to PEGs[3,22]. We found a substantial overlap between the loci previously described as PEGs and reductions of H3K27me3 in *vel3* mutants compared to WT (Fig. 4e)[23]. At well-known PEGs such as *PHE1*, *SUVH7*, *PEG2*, and *VIM5*[1,12], we could clearly show that increased expression is associated with a reduction of H3K27me3 and the presence of the VEL3 protein in early developing seeds (Fig. 4f). Therefore, we conclude that VEL3 is specifically required for the silencing of maternally derived alleles of PEGs, consistent with the role of PRC2 in silencing loci demethylated in the central cell of the female gametophyte in the endosperm.

### *ORE1* plays a role in seed dormancy control

To understand how VEL3 controls seed dormancy we analyzed gene expression changes in mature WT and *vel3-1* endosperms using RNA-seq from samples taken either 1 h (designated the mature stage in this study) or 24 h after imbibition (24HAI). Because VEL3 is a repressor of gene expression we focussed on 124 genes we found with elevated expression in *vel3-1* relative to WT in dry seed endosperms (Supplementary Data 3). GO-term analysis of the upregulated genes indicated a clear enrichment for genes involved in response to stress, reactive oxygen species (ROS) and conditions that cause ROS production (Fig. 5a). Previously it has been shown that H3K27me3 represses ROS-associated programmed cell death pathways activated in response to pathogen infection[24]. We found a considerable overlap between genes mis-regulated in *vel3-1* and pathogen-induced genes repressed by H3K27me3 (Fig. 5b), suggesting that common processes are induced by loss of *VEL3* and pathogen infections (Supplementary Fig. 6a, b). This gene set includes three transcription factors associated with senescence: *ORESERA1* (*ORE1*)[25], *WRKY75*[26] and *NAC046*[27], a close relative of *ORE1*. *ORE1* expression is also repressed by excess H3K27me3 in plants lacking the H3K27me3 demethylase REF6[28] and we found the same is true for *WRKY75* and *NAC046* in the endosperm (Fig. 5c, Supplementary Fig. 6c, d and ref. 11). Interestingly REF6 has already been shown to repress seed dormancy[11,29]. At these loci H3K27me3 is already established during the coenocytic phase of endosperm development when *VEL3* is expressed, and the presence of H3K27me3 at *ORE1*, *WRKY75* and *NAC046* in the mature endosperm requires VEL3 (Fig. 5d). Interestingly the difference in *ORE1* gene expression between WT and *vel3-1* mutants appears at the onset of seed maturation, persisting into maturity and during seed imbibition (Fig. 5e). Thus, the maternal repressive epigenetic state at *ORE1* established in the central cell or early endosperm development antagonizes the later activation of *ORE1* by the seed maturation program. Next we tested whether *ORE1*, *WRKY75* and *NAC046* play a role in seed dormancy control. We analyzed freshly harvested seeds set at 22 °C and found that *ore1-1* mutants[25] showed increased dormancy whereas

*35S:ORE1* seeds[25] showed faster germination than Col-0 (Fig. 5f). Germination of *vel3-1ore-1* double mutants indicates that upregulation of *ORE1* partially accounts for the low dormancy of *vel3-1* seeds when set at 16 °C, whereas *35S:ORE1* did not show a *vel3*-like reduced seed dormancy phenotype, perhaps consistent with previous reports of low activity of the 35S promoter in the endosperm (Fig. 5g). Mutants in *WRKY75* and *NAC046* showed normal seed dormancy (Supplementary Fig. 6e, f). The establishment of ORE1 as a dormancy-regulator adds to the suite of NAC transcription factors controlling seed dormancy in Arabidopsis[30].

### VEL3 associates with histone deacetylase complexes

In addition to their function as PRC2 accessory proteins, VIN3, VRN5 and VEL1 also associate with a histone deacetylase complex containing SIN3-ASSOCIATED PROTEIN 18 (SAP18), SR45 and HDA19[31]. Furthermore, MSI1 also co-purifies with the histone deacetylase HDA19 in its role regulating ABA signaling[32,33]. To detect proteins present with VEL3 in chromatin remodeling complexes we pulled down VEL3-GFP-associated proteins from developing fruits up to 7 days after anthesis and analyzed them via mass spectrometry. This confirmed that VEL3 associates with MSI1 and further revealed that VEL3 associates with a SAP18-containing complex, which putatively included the histone deacetylases HD2C and HDA19 (Supplementary Data 4 and ref. 31). Like VEL3, Arabidopsis HDAC complexes have previously been shown to reside in the nucleolus[34,35]. Using FRET we could confirm that VEL3 interacts with HD2C in the nucleolus in *Nicotiana* leaves (Fig. 6a, b), despite the fact that we could detect no interaction in Y2H assays (Supplementary Fig. 3e). In contrast, HDA19 resided outside of the nucleolus and did not show any direct interaction with VEL3 (Fig. 6a, b). This raised the question of whether VEL3 is required for histone deacetylation in addition to H3K27me3 deposition. Supporting this hypothesis, we found that *ORE1*, *WRKY75* and *NAC046* loci show increased levels of H3K9Ac and H3Ac in *vel3-1* (Fig. 6c; Supplementary Fig. 7). Further supporting the role of H3 deacetylation in seed dormancy control, we found that seeds lacking HD2C show a similar reduced dormancy phenotype to *vel3-1* when set at 16 °C, whereas *hda19-3* mutants showed no phenotype (Fig. 6d), consistent with a recent study[33]. Previously it has been shown that HDA19 regulates the ABA response via overexpression of the ABA biosynthesis genes *NCED5* and *NCED9*[33]. We did indeed find that these genes are overexpressed in *hda19-3* seeds, but not in *hd2c-1* (Fig. 6e). The increased expression of *NCEDs* in *hda19-3* may antagonize the role of *ORE1* in the dormancy control. However, either HD2C or HDA19 were generally important for repression of *ORE1*, *WRKY75* and *NAC046* expression (Fig. 6f) perhaps suggesting some diversity among HDAC complexes associated with VEL3 in seeds.

To further test whether VEL3 plays a role in histone deacetylation we used ChIP-seq for a genome-wide assessment of H3K9ac levels in mature endosperm tissues. Compared to wild-type *vel3-1*

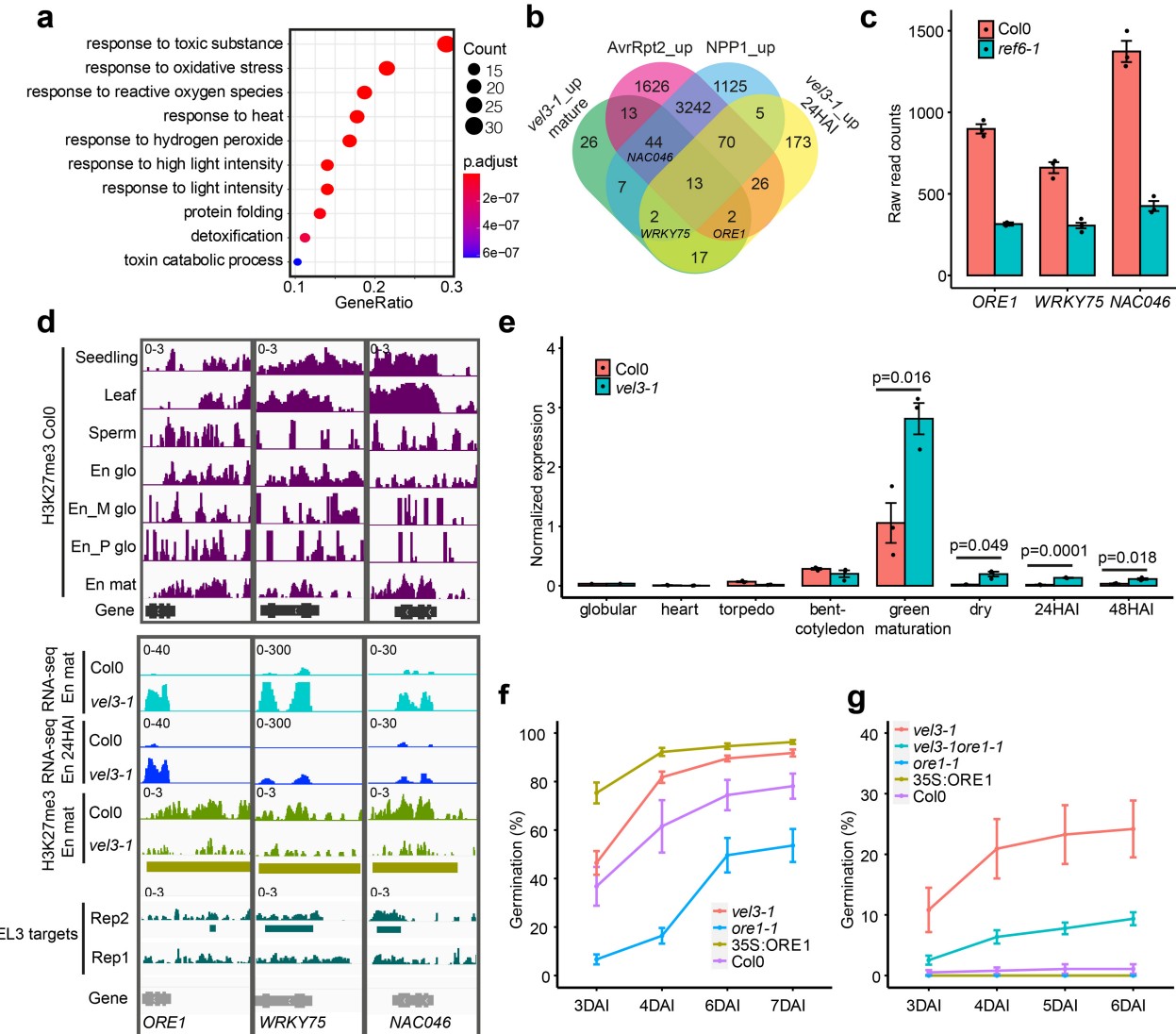

**Fig. 5 | VEL3 is required for the repression of senescence and programmed cell death-associated gene expression in the endosperm with a role in seed dormancy control. a** GO-Term analysis of genes upregulated in mature *vel3-1* seeds compared to Col-0 when set at 16 °C, showing gene function category, *p*-values and number of mis-expressed genes in each category. *p*-values are derived from the one-sided Fisher's exact test. **b** Comparison of upregulated genes in *vel3-1* in the mature endosperm and 24 h after imbibition (24HAI) and genes upregulated by *Pseudomonas syringae* and *Phytophthora parasitica* effector proteins causing programmed cell death, after Tomastikova et al.[24]. **c** Three key senescence-regulating transcription factors are also upregulated by loss of REF6, as shown by RNA-seq in mature Col-0 and *ref6-1* endosperm. Data represent mean and standard error of three biological replicates (data from Sato et al.)[11]. **d** Analysis of the role of VEL3 and H3K27me3 in the regulation of *ORE1*, *WRKY75* and *NACO46* in seeds set at 16 °C. Levels of H3K27me3 are shown at in different tissues, including the maternally derived and paternally derived alleles in the endosperm (Moreno Romero et al.[3];

En_M glo and En_P glo), alongside gene expression levels in mature seeds, 24 h imbibed seeds (24HAI) derived by RNA-seq. En: endosperm; glo: globular; mat: mature. Golden bars indicate significant regions (*p* < 0.05) identified by DiffBind for H3K27me3 between Col0 and *vel3-1*. VEL3 target regions were identified by MACS2. **e** Comparison of *ORE1* gene expression during seed development and imbibition in WT and *vel3-1* seeds set at 16 °C, by qPCR. Data represent the mean +/−SE of 3 biological replicates with significant differences shown by 2-tailed *t*-test at each timepoint. **f** Germination of *ore1-2* and *35 S:ORE1* seeds set at 22 °C. Data represent mean +/− standard error of 6 to 8 biological replicate seed batches. Significant differences were calculated by 1-way ANOVA with Tukey post hoc test. **g** Upregulation of *ORE1* is partially accounts for the low dormancy of *vel3-1* seeds when set at 16 °C. Germination of WT Col-0, *vel3-1*, *ore1-1*, 35S:ORE1, and *vel3-1 ore1-1* double mutants. Data represent mean +/− SE of 8 biological replicate seed batches per genotype. Statistical analysis is via 1-way ANOVA with Tukey post hoc test.

endosperms contained large number of differentially acetylated regions with many loci with increased or decreased H3K9ac (Fig. 7a, b). The large number of hypoacetylated regions was initially a surprise but we found only hyperacetylated regions are targeted by VEL3 (Fig. 7c). HypoH3K27me3 regions and TEs are also overlapped with hyperacetylated regions but not hypoacetylated regions (Fig. 7d, e). Analysis the distribution of hypo- and hyperacetylated regions in *vel3-1* endosperms found that *vel3-1* mutants are characterized by increased H3K9ac in pericentromeric regions (Fig. 7a), coupled with decreased H3K9ac elsewhere in the genome. This suggests that in the absence of

VEL3, HDAC complexes lose their targeting to pericentromeric regions and instead are recruited by other factors elsewhere in endospermic chromatin.

## Discussion

We propose that VEL3 associates with a nucleolar HDAC complex in central cell to establish a trans-generationally inherited repressive epigenetic state at *ORE1* and other genes with a role in suppressing PCD and inducing seed dormancy (Fig. 8), which is required prior to H3K27me3 deposition. *VEL3* is expressed in the central cell and in early

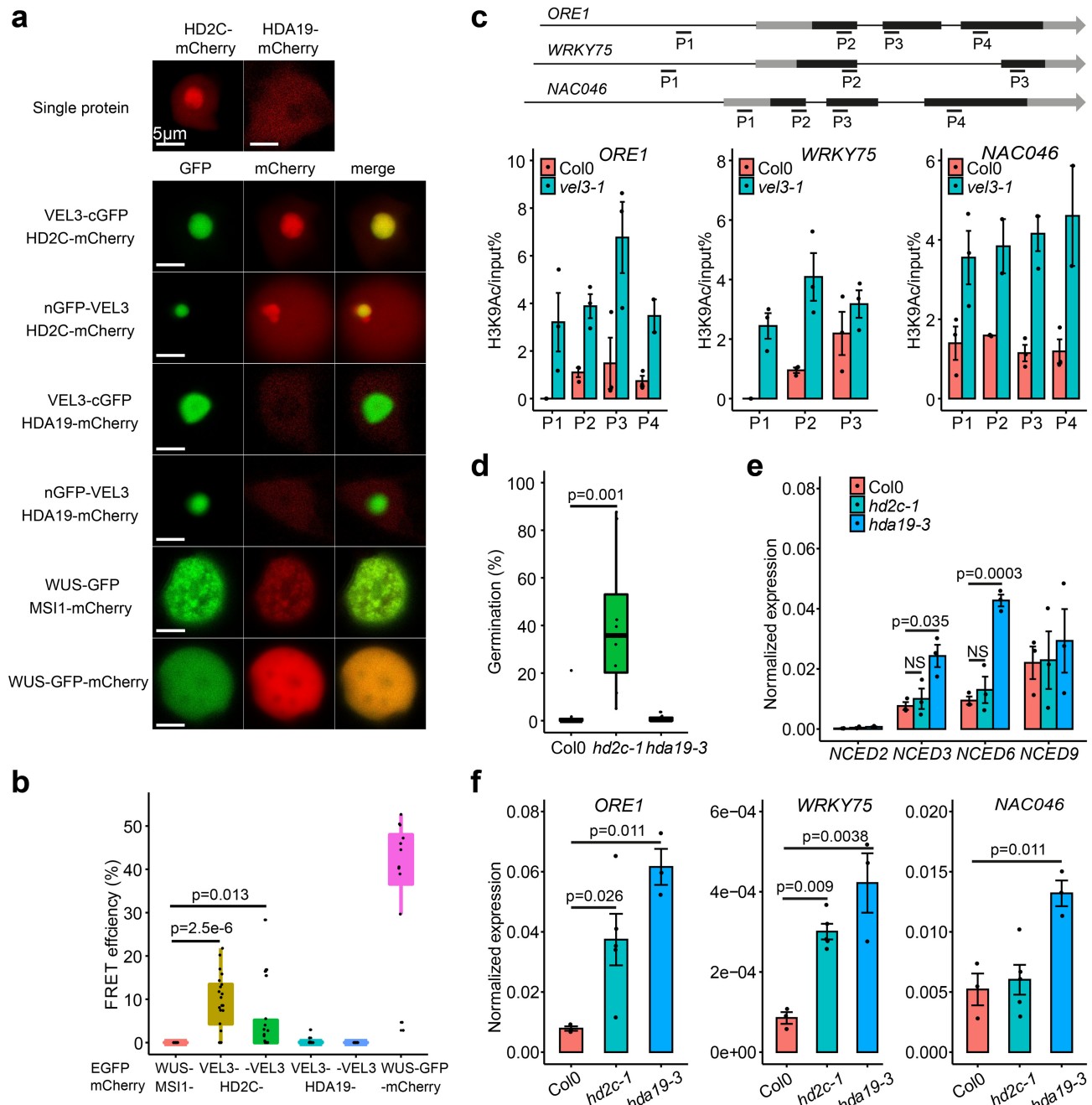

**Fig. 6 | VEL3 is required for the function of histone deacetylase complexes in the endosperm regulating seed dormancy. a** VEL3 co-localizes with HD2C in the nucleolus, but not with HDA19, which exists pre-dominantly in the nucleoplasm during transient expression in *Nicotiana*. **b** FRET assays show that VEL3 and HD2C associate *in planta*. VEL3-: VEL3 C-terminal GFP fusion. -VEL3: VEL3 N-terminal GFP fusion. Self-dimerizing WUS-GFP/mCHERRY is shown as the positive control. Box-plots indicate minimum and maximum values, as well as 25th, 50th, and 75th quartiles. *p*-values are derived from the two-sided *t*-test. Sample sizes are included in source data (*N* = 16,21,18,16,16,12 from left to right). The experiments were repeated twice. **c** VEL3 is required for H3K9 deacetylation at *ORE1*, *WRKY75* and *NAC046*. Gene models show location of exons in black and primers used for ChIP. Charts show H3K9Ac levels in mature endosperms of seeds set at 16 °C. Data shown

as mean +/− SE of three biological replicate immunoprecipitations tested by qPCR. **d** Loss of HD2C causes a similar effect on seed dormancy to loss of VEL3 in seeds set at 16 °C, but *hda19-3* mutants show a WT phenotype. Data represents mean +/− SE of 8 to 12 biological replicate seed batches per genotype. Boxplots indicate minimum and maximum values, as well as 25th, 50th, and 75th quartiles. *p*-values are derived from the two-sided *t*-test. **e** Analysis of *NCED* gene expression in WT, *hd2c-1* and *hda19-3* mutants set at 16 °C. Data shown as mean +/− SE of three biological replicates tested by qRT-PCR. *p*-values are derived from the two-sided *t*-test. **f** qRT-PCR analysis of the expression of *ORE1*, *WRKY75* and *NAC046* in *hd2c-1* and *hda19-3* mutants. Data shown as mean +/− SE of three biological replicates. *p*-values are derived from the two-sided *t*-test.

endosperm development and the strong maternal bias to the effect of VEL3 on seed dormancy suggests a function in the central cell or very early post-fertilization. In agreement with this hypothesis VEL3 appears to target loci previously described to encode PEGs in wild-type seeds (Fig. 4e). Further support for this hypotheses comes from the

observation that the distribution of H3K27me3 in the mature endosperm is already established in coenocytic stage of endosperm development (Fig. 4c)[21]. This suggests that VEL3 creates maternal primed epigenetic marks functioning through to and after seed dispersal to increase seed dormancy.

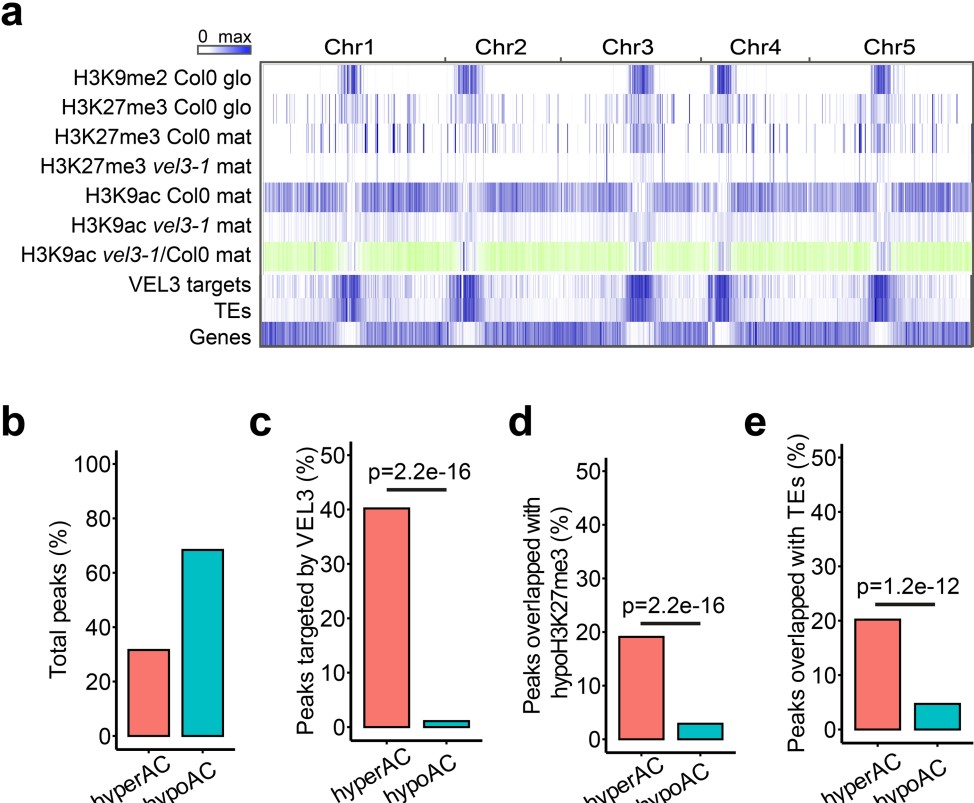

**Fig. 7 | VEL3 is required for H3K9Ac to pericentromeric regions. a** Genome-wide distribution of H3K9me2, H3K27me3, and H3K9ac in Col-0 and *vel3-1* (glo-globular embryo stage; mat- mature endosperm. Blue color indicates the regions marked by the indicated epigenetic modification, VEL3-direct targets by ChIP-seq and the presence of transposable elements (TEs). For H3K9ac in *vel3-1* vs. Col-0, blue indicates significantly hyperacetylated regions, green hypoacetylated regions.

**b** Percentage of total hypo- and hyperacetylated peaks in *vel3-1* vs. Col-0 mature endosperm. **c** Percentage of differentially acetylated peaks bound by VEL3-GFP in early endosperm development. **d** Percentage of differentially acetylated peaks marked by H3K27me3 in mature endosperms. **e** Percentage of differentially acetylated peaks in transposable elements (TEs). Statistical significance was determined using Chi-Square test in **c**–**e**.

While VEL3 is required maternally to distribute repressive marks in the endosperm, the effect of VEL3 on gene expression controlling dormancy does not appear until seed maturation (Fig. 5d, e). This period is when seed dormancy is established, and suggests that the maturation program does not require the wholesale epigenetic reprogramming of the endosperm, for instance by the transcription factors known to associate with PRC2[31,36–38]. Instead the mother plant appears to control the gene expression level of dormancy-related genes during seed maturation and germination by creating a primed repressive epigenetic state in the female gametophyte or shortly after fertilization.

VEL3 associates with the HDAC complex and MSI1, as has been described for VRN5. VEL3 can recruit MSI1 to the nucleolus, which is the site of HDAC activity[34,35,39], and VEL3 is required for normal recruitment of HDAC activity to pericentromeric regions (Fig. 7). One possibility is that VEL3 recruits MSI1 to the nucleolar HDAC complexes with PRC2 activity and H3K27me3 deposition contingent on prior deacetylation rather than a requiring VEL3 per se. Alternatively VEL3-MSI1 complexes may relocate to the nucleoplasm after H3 deacetylation with or without bound chromatin to participate directly in H3K27me3 deposition. Indeed a recent study shows that a similar HDA19-MSI complex in germinating seeds establishes the histone deacetylation state prior to H3K27me3 deposition[33]. Histone deacetylase HDA9 has also been suggested to be required for the PRC2 activity in the repression of *FLC*[40]. Notably VRN5 and PRC2 and HDAC are required for repression of the ABA response in seedlings[32,41,42]. However, a lack of *VEL3* expression at this stage confines VEL3 to the control of primary seed dormancy, which is established during seed development. Interestingly, there is a strong overlap between PRC2-regulated genes important for programmed cell death, ABA induced senescence and seed dormancy: indeed processes governing plastid reprogramming and de-greening are also shared between senescent tissues and dormant tissues.

## Methods

### Plant materials and growth conditions

All *Arabidopsis thaliana* natural accessions used in this study are listed in Supplementary Data 1. T-DNA mutants were obtained from NASC (UK): *vel3-1* (SALK_052041), *vel3-2* (FLAG_383F07), *hd2c-1* (SALK_129799C)[42], *hda19-3* (SALK_139445)[43], *ref6-1* (SALK_001018), *ore1-1* (SALK_090154C)[25], *wrky75-25* (N121525)[26], *nac046-1* (SK2690)[27]. 35S:ORE1 was a kind gift from Salma Balazadeh[25]. All mutants are in Col-0 background except *vel3-2*, which is in WS background. For *vel2* we obtained seeds for lines SALK_205136, SALK_033362, SALK_003461, SALK_033363, SALK_127770 and SALK_136413 but did not obtain a heterozygous or homozygous plant for any line despite testing up to 20 individuals for each line. Other mutant lines used were: anac060-1, SALK_12554C; *anac060-2*, SALK_127838C; *erd6-1*, SALK_132824C; *erd6-2*, SALK_083717C; *rd22-1*, SALK_146066C; *rd22-2*, SALK_122821C; *at4g10290-1*, SALK_010871C; *at4g10290-2*, SALK_38425C.

Two different versions of VEL3 complementary lines were generated by floral dipping with *Agrobacterium tumefaciens*, given that the TAIR10-annotated VEL3 could not completely rescue the phenotype of *vel3*. For VEL3.1-re-seq version, VEL3 promoter (1.4 kb), genomic region (1 kb) and 3′ (1 kb) was amplified by PCR, then assembled using Golden

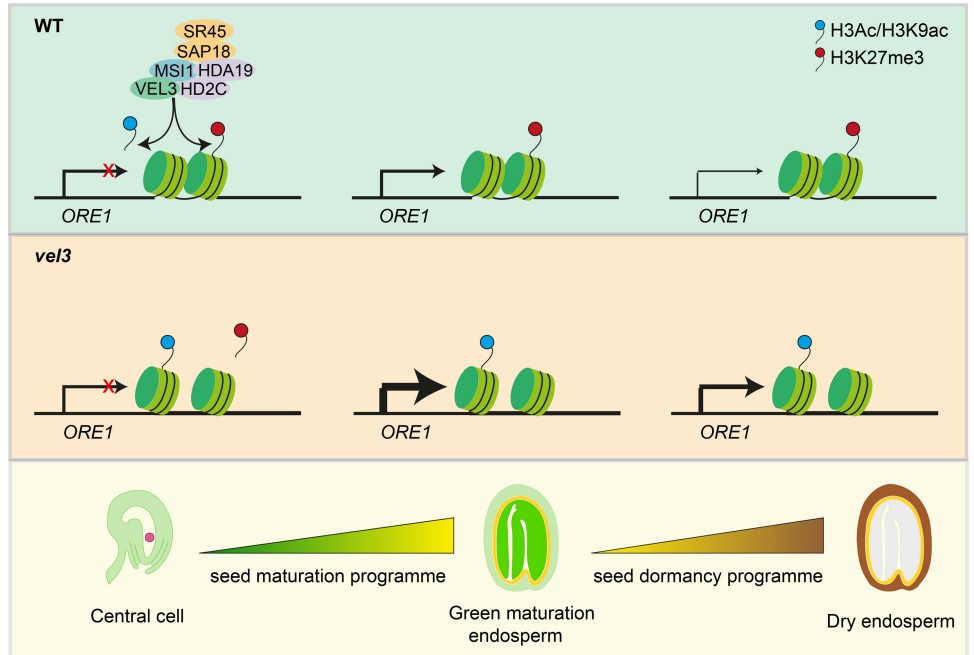

**Fig. 8 | VEL3 associates with the HDAC complex and possibly also PRC2 in the female gametophyte to establish a maternal repressive epigenetic state on dormancy-associated genes (e.g., ORE1).** The distribution of H3K27me3 is stable and determines the expression level of targeted genes once they are activated during the maturation stage and during imbibition. At the stages of green maturation and dry seeds, in *vel3* mutant thicker lines indicate stronger transcription of ORE1 due to the less condensed chromatin.

Gate with or without eGFP. For VEL3.2-TAIR10 version, VEL3 coding sequence according to the TAIR10 was synthesized by Thermo Fisher scientific. The VEL3 promoter (1.4 kb), synthesized CDS (0.72 kb) and 3′(1 kb) were assembled using Golden Gate. The constructs were transformed into *vel3-1* plants.

For the genome-wide association study, *Arabidopsis* natural accessions were vernalized for 6 weeks to induce flowering before growing at 22 °C or 16 °C under white light at 80–100 µmol m² s⁻¹ in long days (16 h light). For all the other experiments, plants were directly grown in long days without vernalization (16 h light at 22 °C, 8 h darkness at 20 °C). Then plants were either maintained in 22 °C day/20 °C night or transferred to 16 °C day/14 °C night at first flowering. All mutants and wildtype were grown side-by-side in different experiments for solid comparison. Matured seeds were freshly harvested for the seed dormancy assay.

### Seed dormancy assay
For seed production plants were grown in 16 h light/8 h dark for seed dormancy assays at either 16 °C or 22 °C as indicated. Seed dormancy assay was performed in 0.9% water agar plates. Seed batches from individual mother plants were used as biological replicates. For GWAS, up to 5 biological replicates of 30-50 seeds from independent mother plants were used for each accession and treatment. For the other experiments, at least 6 biological replicates of ~100 seeds were used. Seed germination was scored by radicle emergence at 7 days after sowing. All statistical analyses in this study were performed using R (http://www.R-project.org/). For multiple tests, two-way ANOVA with post hoc test was applied using a threshold of $p$-value < 0.05. All the other tests were performed using two-tailed Student's $t$-test. The %germination data was arcsine transformed prior to statistical analysis.

### Genome-wide association study (GWAS)
*Arabidopsis* accessions were obtained from the European Arabidopsis Stock Centre (NASC). GWAS was performed using the non-parametric Kruskal–Wallis approach, implemented in the GWA-Portal[13].

### Haplotype analysis
SNPs in the phyB/VEL region from 8140285 to 8179319 were obtained from Horton et al.[44], and two further SNPs from Seren[13] significantly associated with dormancy were added to give a total of 99 SNPs across 213 accessions. The SNPs from each accession were used to construct an input file for fastPHASE version 1.4.8[45]. fastPHASE was run as:./fastPHASE -T1 -o Output -H200 -Pzp -K3 InputFile.inp. The -Pzp option outputs expected total cluster memberships of each SNP per accession. The haplotype cluster membership with the highest probability was designated as the haplotype for that SNP.

### Yeast two hybrids (Y2H)
Sequences were amplified from globular stage seeds, using primers with the appropriate Gateway sites (see Table S5 for primer details) and cloned into BP Gateway pDONR221 Vector (Invitrogen), then LR recombined into pDEST22 (prey) and pDEST32 (bait) (both Invitrogen), for fusion to the GAL4 DNA binding domain (BD) or activation domain (AD), and subsequently transformed into yeast cells strain AH109. Transformed yeast cells were spotted on stringent selective synthetic defined (SD) media lacking Trp (W), Leu (L), His (H), and adenine (A) (SD-WLHA), or SD-WLH and SD-WLH containing various concentrations, 1 to 10 mM, of 3-Amino-1,2,4-Triazol (3AT) (SD-WLH + 3AT). Yeast cells were also spotted on SD media lacking W and L (SD-WL) to check for successful transformation with the plasmids.

### Fluorescence resonance energy transfer (FRET)
FRET was performed in transient transfection assay on *Nicotiana benthamiana* leaves. Coding sequences of VEL3, MSI1, HD2C and HDA19 were amplified, and assembled with 35S promoter module, eGFP or mCherry module using Golden Gate. *Agrobacterium tumefaciens* GV3101 carrying the desired constructs was cultured overnight and resuspended in 0.01 M 2-(N-morpholino) ethanesulfonic acid (MES) pH 5.6, 0.01 M MgCl₂, 0.01 M acetosyringone. Each *Agrobacterium* strain carrying the desired constructs was syringe infiltrated into leaves of 4-week-old *Nicotiana benthamiana* plants. Material for

experiments was harvested and imaged 3 days post-infiltration. 5 mm pieces of leaf were cut, mounted in water and imaged with a Leica 63X water immersion objective. Acceptor photobleaching FRET analysis was carried out using the Leica SP8X confocal microscope. In brief, 20 multi-scanned images for eGFP and mCherry proteins were collected under donor excitation wavelength (488 nm) and acceptor excitation wavelength (561 nm), respectively. Images were acquired using the laser power set at 0.2% maximum to minimize photobleaching. The mCherry was photobleached by continuously scanning region of interest (ROI) with the 561-nm laser line set at 100. Recovery of the donor from quenching was quantified by subtracting the post-bleaching donor emission from that of pre-bleaching. The FRET efficiencies were calculated after automatic background subtraction from manually defined background ROIs. Acquisition and analysis of FRET images were conducted using the Leica Application Suite LAS AF.

## Microscopy

DIC images of developing seeds were taken as previously described[4]. Siliques were cut in half, fixed with EtOH:acetic acid (9:1), washed for 10 min in 90% EtOH, 10 min in 70% EtOH and cleared in chloral hydrate:glycerol:H2O solution (8:1:3). Seeds were observed under differential interference contrast (DIC) optics using a Zeiss Axio Imager Z2 microscopes (Carl Zeiss AB, Sweden). For the fluorescence microscopy tissues were dissected and mounted in PBS buffer. Samples were counterstained with DAPI and analyzed under confocal microscopy on ZEISS LSM880 Airiscan or Leica SP8X. For Feulgen staining, whole siliques were fixed in an EtOH:acetic acid (3:1) solution overnight. Before staining, siliques were washed three times with distilled water for 15 min and then immersed in 5 N HCl for 60 min. Following hydrolysis, siliques were washed three times with cold distilled water for 10 min. Siliques were stained for 2 h in the dark with Schiff's reagent and then washed three times in distilled water for 10 min. Siliques were dehydrated in an ethanol series (50%, 70%, 95%, and 100%). Every hour, 100% ethanol was changed until the siliques stopped turning pink. Then, whole siliques were embedded in ethanol:Technovit® 7100 resin (1:1) for 2 h. The samples were then incubated overnight in Technovit® 7100, and seeds were dissected out of the siliques and mounted in Technovit® 7100 plus accelerator for 2 h at room temperature for polymerization. The seeds were imaged using a Leica Stellaris 8 FAL-CON confocal microscope with a white light laser and excitation at 488 nm and the detector at 535 nm and longer.

## RNA-seq and qPCR

To extract RNA from globular seeds, from plants grown at 16 °C, at 8 days after pollination, seeds from 3-5 siliques were manually dissected and pooled as one biological replicate. RNA was extracted using RNeasy Plant Mini Kit (Qiagen, 74904). To extract RNA from mature and imbibed endosperm, 0.05 g seeds were imbibed for 1 (mature) or 24 h (24HAI), prior to purification of endosperm-enriched tissues as previously described[46]. In brief, seeds were squeezed with glass slides after imbibition. Endosperm-enriched tissues were centrifuged at maximum speed twice with 40% sucrose, washed three times with water, before stored in −80 °C for further RNA extraction. RNA was extracted using the borate method[47]. Ground endosperm tissues were incubated in 250 µl borate extraction buffer (0.2 M sodium borate decahydrate, 30 mM EGTA, 1% SDS, 1% sodium deoxycholate) and 10 µl proteinase K, at 42 °C for 90 min. 20 µl 2 M KCl was added, incubated on ice for 60 min. Samples were centrifuged at 13,000 rpm, 4 °C for 20 min to remove debris. Supernatant were collected to a fresh tube and added 90 µl of 8 M LiCl. RNA was precipitated in −20 °C overnight, before collected by spinning down at 13,000 rpm, 4 °C for 20 min. RNA pellet was dissolved in water and purified with Qiagen RNeasy Plant Mini Kit (Qiagen, 74904). On-column DNase I digestion (Qiagen, 79254) was performed during RNA extraction. Three biological replicates were prepared. For RNA-seq, RNA samples were sequenced at

Novogene (Beijing, China) using a Hiseq 4000 system, 150 bp paired-end sequences with a minimum of 25 million reads acquired per sample. For qPCR, cDNA samples were prepared using the Quantitect reverse transcription kit (Qiagen, 205311). qPCR was then performed with the SYBR Green master mix (Bio-Rad, 1725271). *ACT2* was used as the reference gene.

## ChIP-seq and ChIP-qPCR

For H3K27me3 ChIP-seq, at maturation stage, endosperm of Col0 and *vel3-1* was purified from 0.5 g seeds after 1 h imbibition as previously described (see above method RNA-seq and qPCR section). Endosperm tissue were cross-linked in 1% formaldehyde solution for 15 min and quenched with 0.125 M glycine for 5 min. Then ChIP was performed as previously described[48]. In brief, tissues were ground in liquid nitrogen, resuspended in 30 ml extraction buffer1 (0.4 M sucrose, 10 mM Tris−HCl, 5 mM βME, protease inhibitor cocktail), filtered through two layers of Miracloth. The filtered solution was centrifuged for 20 min at 2880 × $g$ at 4 °C. Pellet was dissolved in 1 ml extraction buffer 2 (0.25 M sucrose, 10 mM Tris−HCl, pH 7.5, 10 mM MgCl2, 1% Triton X-100, 5 mM βME, protease inhibitor cocktail), centrifuged at 12,000 × $g$ for 10 min at 4 °C. Pellet was resuspended in 300 µl extraction buffer 3 (1.7 M sucrose, 10 mM Tris−HCl, pH 7.5, 0.15% Triton X-100, 2 mM MgCl2, 5 mM βME, protease inhibitor cocktail), overlayed by additional 90 µl extraction buffer 3, centrifuged at 16,000 × $g$ for 1 h at 4 °C. Pellet was resuspended in 320 µl nuclei lysis buffer (50 mM Tris−HCl, pH 7.5, 10 mM EDTA, 1% SDS, protease inhibitor cocktail), sonicated to 200−1000 bp fragments. The sonicated chromatin solution was incubated overnight at 4 °C, with 25 µl Dynabeads protein A magnetic beads (ThermoFisher, 10001D) and 5 µl anti-H3K27me3 antibody (Merk Millipore, 07-449). After immunoprecipitation, IP samples were washed with low-salt wash buffer (150 mM NaCl, 0.1% SDS, 1% Triton X-100, 2 mM EDTA, 20 mM Tris−HCl, pH 7.5), high salt wash buffer (500 mM NaCl, 0.1% SDS, 1% Triton X-100, 2 mM EDTA, 20 mM Tris−HCl, pH 7.5), LiCl wash buffer (0.25 M LiCl, 1% NP-40, 1% sodium deoxycholate, 1 mM EDTA, 10 mM Tris−HCl, pH 7.5), and TE buffer (10 mM Tris−HCl, pH 7.5, 1 mM EDTA), resuspended in elution buffer (1% SDS, 0.1 M NaHCO3). DNA was purified using iPure kit v2 (C03010015). 5 ng ChIP-ed DNA and input DNA were sent to BGI (China) for sequencing, in the platform of BGISEQ, 50 bp single-end with a 20 M clean reads per sample. Two biological replicates were sequenced.

For H3K9ac ChIP-seq, immunoprecipitation was performed similarly as H3K27me3, except that 1.5 g mature seeds were used. DNA was extracted using phenol-chloroform method. 20 ng ChIP-ed DNA and input DNA were sent to Novogene (China) for sequencing, 150 bp paired-end with a minimum25M clean reads per sample. Anti-H3K9ac (Merck Millipore, 07-352) was used in the immunoprecipitation. Two biological replicates were sequenced.

For VEL3 ChIP-seq, two gram *vel3-1 gVEL3-GFP* siliques from globular stage were collected. Siliques were cut into pieces before cross-linking in 1% formaldehyde solution for 20 min and quenched with 0.2 M glycine for 6.5 min. Nuclei were extracted according to the previously described eChIP-seq[49]. 0.2 g ground materials were lysed in 300 µl Buffer S (50 mM HEPES-KOH, pH 7.5, 150 mM NaCl, 1 mM Ethylene Diamine EDTA, 1% Triton X-100, 0.1% sodium deoxycholate, 1% SDS) for 10 min at 4 °C, then mixed with 1.2 ml Buffer F (50 mM HEPES-KOH, pH 7.5, 150 mM NaCl, 1 mM EDTA, 1% Triton X-100, 0.1% sodium deoxycholate). The remaining sonication and immunoprecipitation steps are same as above mentioned H3K27me3 ChIP-seq. DNA was then purified using iPure kit v2 (C03010015). Tn5 tagmentation was performed on 1 ng ChIP-ed DNA or input DNA for 5 min at 55 °C, with 0.1 µl enzyme and 2.5 µl buffer in 50 µl reaction (Illumina, 20034197). PCR amplification was conducted using the following programs: 72 °C 5 min, 98 °C 30 s, N cycles of 98 °C 10 s, 63 °C 30 s and 72 °C 1 min. Cycle number (N) was determined by the qPCR of each

partially amplified library. 0.8X Agencourt AMPure XP beads (Beckman Coulter, A63880) was used for the library size selection. Libraries were sequenced at Novogene (Beijing, China) using a Novaseq system, 150-bp paired-end sequences with a minimum of 30 million reads acquired per sample. GFP–Trap magnetic agarose beads (25 µl) (Chromotek, GTMA-20) were used in the immunoprecipitation. Input was served as control for each sample. Two biological replicates were sequenced. For ChIP-qPCR in mature endosperm, procedure was same as H3K27me3 ChIP-seq (see above), except 1 g seeds were used. Anti-H3ac (Merck Millipore, 06-599) and Anti-H3K9ac (Merck Millipore, 07-352) were used in the immunoprecipitation. Three biological replicates were used. Final IP efficiency was normalized to the input percentage.

## RNA-seq data analysis
Raw RNA-seq reads were trimmed using cutadapt-1.9.1[50] and mapped to *Arabidopsis thaliana* TAIR10 reference genome using STAR-2.5.a[51]. featureCounts[52] was used to count the numbers of reads mapped to each gene. Sense reads were selected for downstream analysis. edgeR[53] was used to calculate differentially expressed genes (DEGs), based on the threshold of FDR < 0.05 and fold-change ≥ 2. For data visualization, bigwig files were generated using deepTools-3.1.1[54] with a bin size of 50 bp, before visualization in IGV −2.12.3[55].

## ChIP-seq data analysis
Raw reads were trimmed using cutadapt-1.9.1[50] and mapped to *Arabidopsis thaliana* TAIR10 reference genome using bowtie2[56]. H3K9ac ChIP-seq datasets were subsampled to the same read depth (25 M) for each sample. Only uniquely mapped reads were kept for downstream analysis using Samtools-1.9 and Sambamba-6.7[57,58]. The reproducibility of biological replicates were assessed by computing pairwise Pearson's correlation coefficients (PCCs), using multiBamSummary in deepTools-3.1.1[54], with the bin size of 10 kb. PCC was >0.9 for all ChIP-seq replicates. For data visualization, bigwig files were calculated using deepTools-3.1.1[54] with a bin size of 50 bp, before visualization in IGV −2.12.3[55]. Peaks were called using SICER 1.1 for H3K27me3[59] or MACS2 for H3K9ac and VEL3[60], based on the threshold of FDR < 0.05. Only common peaks between two biological replicates were considered. Differentially marked genes were calculated using DiffBind[61] and DESeq2[62], based on the threshold of FDR < 0.05 and fold-change ≥ 2. Genes were identified for these peaks are located in genebody or 1 kb promoter using bedtools-2.28.0[63]. Previously published H3K27me3 data in different tissues were re-analyzed using the same pipeline. Seedling data is from ref. 64; leaf data is from ref. 65; sperm data is from ref. 23; globular endosperm data is from ref. 3.

## Immunoprecipitation-mass spectrometry (IP-MS)
IP-MS was performed as previously described[66]. Four grams of *vel3-1 gVEL3-GFP* siliques from globular stage were collected as one biological replicate. 35S-GFP siliques were served as control. Two biological replicates were used. Siliques were cut in half, cross-linked in 1% formaldehyde solution for 15 min and quenched with 0.125 M glycine for 5 min. Cross-linked tissues were ground in liquid nitrogen and lysed in 30 ml cell lysis buffer(20 mM pH 7.5 Tris–HCl, 250 mM sucrose, 25% glycerol, 20 mM KCl, 2.5 mM MgCl$_2$, 0.1% NP-40, 5 mM DTT), filtered through two layers of Miracloth, centrifuged for 15 min at 4000 rpm at 4 °C. The pellets were washed twice with 1 ml of nuclear wash buffer (20 mM Tris–HCl, pH 7.5, 2.5 mM MgCl$_2$, 25% glycerol, 0.3% Triton X-100, 5 mM DTT) and resuspended with 1.2 ml nuclei lysis buffer (1× PBS, 1% NP-40, 0.5% sodium deoxycholate, 0.1% SDS). The chromatin was sonicated to 200–1000 bp, and pelleted at 13,000 × *g* at 4 °C for 15 min. Roche Complete EDTA-free protease inhibitor cocktail was added to all buffer (Roche,11873580001). Co-IP was performed using GFP–Trap magnetic agarose beads (25 µl per sample) (Chromotek, GTMA-20), at 4 °C for 2 h. Magnetically separated beads were washed

twice with 500 µl High-salt wash buffer (20 mM Tris–HCl, pH 7.5, 500 mM NaCl, 0.5 mM EDTA, 0.1% SDS and 1% Triton X-100), and twice with 500 µl low-salt wash buffer (20 mM Tris–HCl, pH 7.5, 150 mM NaCl, 0.5 mM EDTA, 0.1% SDS and 1% Triton X-100), eluted by heating at 95 °C 15 min in 1x SDS loading buffer.

The protein samples were purified from 10% SDS−PAGE gels for in-gel trypsin digestion using standard procedures adapted from Shevchenko et al.[67]. Briefly, the slices were washed with 50 mM TEAB buffer pH 8 (Sigma), incubated with 10 mM DTT for 30 min at 65 °C followed by incubation with 30 mM iodoacetamide (IAA) at room temperature (both in 50 mM TEAB). After washing and dehydration with acetonitrile, the gels were soaked with 50 mM TEAB containing 10 ng/µl Sequencing Grade Trypsin (Promega) and incubated at 40 °C for 8 h. The peptides were eluted with an equal volume of 5% formic acid followed by different steps of acetonitrile concentration (up to 50%). The combined supernatants were dried in a SpeedVac concentrator (Thermo Fisher Scientific, #SPD120) and the peptides dissolved in 0.1% TFA/3% acetonitrile.

Aliquots were analyzed by nanoLC-MS/MS on an Orbitrap Eclipse™ Tribrid™ mass spectrometer coupled to an UltiMate® 3000 RSLCnano LC system (Thermo Fisher Scientific, Hemel Hempstead, UK). The samples were loaded and trapped using a pre-column with 0.1% TFA at 15 µl min$^{-1}$ for 3 min. The trap column was then switched in-line with the analytical column (nanoEase M/Z column, HSS C18 T3, 100 Å, 1.8 µm; Waters, Wilmslow, UK) for separation using the following gradient of solvents A (water, 0.1% formic acid) and B (80% acetonitrile, 0.1% formic acid) at a flow rate of 0.2 µl min$^{-1}$: 0–3 min 3% B (during trapping); 3–10 min linear increase B to 9%; 10–70 min increase B to 40%; 70–90 min increase B to 60%; followed by a ramp to 99% B and re-equilibration to 3% B, for a total running time of 122 min. Mass spectrometry data were acquired with the following MS settings in positive ion mode: MS1/OT: resolution 120 K, profile mode, mass range *m/z* 300−1800, spray voltage 2800 V, AGC 4e5, maximum injection time of 50 ms; MS2/IT: data dependent analysis was performed using parallel HCD and CID fragmentation with the following parameters: top20 in IT turbo, centroid mode, isolation window 1.0 Da, charge states 2-5, threshold 1.0e4, CE = 33, AGC target 1.0e4, max. inject time 35 ms, dynamic exclusion 1 count, 15 s exclusion, exclusion mass window ±10 ppm.

Peaklists (mgf) generated from the raw files using the msconvert tool from Proteowizard[68] were used for the database search for peptide and protein identification using an in-house Mascot Server 2.7.0.1 (Matrixscience, London, UK). Mascot was set up to search the TAIR10_pep_20101214 Arabidopsis thaliana protein sequence database (arabidopsis.org, 35,386 entries) plus the Maxquant contaminants database (245 entries) with the following parameters: enzyme trypsin, 2 missed cleavages, oxidation (M), deamidation (N,Q) and acetylation (protein N-term) as variable and carbamidomethylation (C) as fixed modifications, precursor tolerance 6 ppm, fragment tolerance 0.6 Da. The Mascot search results were imported into Scaffold 4.11.0 (www.proteomsoftware.com) using identification probabilities of 99% for proteins and 95% for peptides. For quantitative analysis the exclusive unique spectra counts were exported from Scaffold into Microsoft Excel. The significance of differences between VEL3 and control IPs were evaluated using the SAINTexpress tool for Significance Analysis of interactomes[69]. The Bayesian false discovery rate (BFDR) and fold-change were provided in Supplementary Data 4. The mass spectrometry proteomics data have been deposited to the ProteomeXchange Consortium via the PRIDE[70] partner repository with the dataset identifier PXD041094.

## Quantification and statistical analysis
Quantification of germinated seeds was determined by radicle emergence at 7 days after sowing. Quantification of abnormal seeds was carried out under microscope using mature seeds.

All statistical analyses in this study were performed using R (http://www.R-project.org/). For multiple tests, one-way ANOVA with post hoc test was applied using a threshold of $p$-value < 0.05. All the other tests were performed using two-tailed Student's $t$-test. The % germination data was arcsine transformed prior to statistical analysis.

## Reporting summary

Further information on research design is available in the Nature Portfolio Reporting Summary linked to this article.

## Data availability

All the RNA-seq and ChIP-seq data generated from this study have been deposited in the Gene Expression Omnibus under accession code GSE202802. Raw IP-MS data have been deposited to the ProteomeXchange Consortium via the PRIDE partner repository with the dataset identifier PXD041094. The unique biological materials are available upon appropriate requests. Source data are provided with this paper.

## Code availability

Custom code used in this study are available upon request.

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

## Acknowledgements

The authors would like to thank Dr. S. Balazadeh (Leiden University, Leiden) for the 35 S:ORE1 line. We are also grateful to Dr. C. Faulkner (John Innes Centre, Norwich) and Dr. Y. Stahl (Heinrich Heine University, Dusseldorf) for the WUS:GFP and WUS:GFP:mCherry constructs. We thank Nick Pullen (John Innes Centre, Norwich) for the contribution on haplotype analysis. We also thank W. Huang, P. Zhu, H. Gao and S. Zhou for the technical assistance, and C. Xu for the discussion. The proteomics analysis was performed by Carlo Martins and Gerhard Saalbach at the Proteomics Facility of the John Innes Centre, Norwich, UK, supported by the BBSRC core capability grant.

## Author contributions

S.P. conceived the project; X.C. conducted experiments and data analysis; D.R.M. and N.Z. performed the GWAS and initial mutant characterization; F.S. performed the Y2H and FRET; X.C. performed the bioinformatics; T.B.G. contributed to the methodology and discussion; S.P. wrote the initial manuscript; All authors edited the manuscript.

## Competing interests

The authors declare no competing interests.

## Additional information

**Peer review information** : *Nature Communications* thanks the anonymous reviewer(s) for their contribution to the peer review of this work. Peer reviewer reports are available.

