## [Peer Review File · Nature Communications]

A VEL3 histone deacetylase complex establishes a maternal epigenetic state controlling progeny seed dormancyReviewer #1 (Remarks to the Author):

This manuscript reported that Arabidopsis VERNALIZATION5/VIN3-LIKE 3 (VEL3) maintains maternal control over progeny seed dormancy by establishing an epigenetic state of histone methylation and acetylation early in endosperm development. VEL3 associates with PRC2 and HDAC complex to establish a trans-generationally inherited repressive epigenetic state. The experiments were well designed and results are interesting. However, I am concerning the novelty of this manuscript.

1) It is not surprised that VEL3 is required for the establishment of H3K27me3 deposition did not give me surprise, which did not highlight the novelty of this work. Five-member VEL gene family was observed in the Arabidopsis genome and VEL1 has been reported acting as component in PHD-PRC2 complex to trigger the epigenetic silencing of FLC during vernalization (Filomena De Lucia et al., 2008, PNAS). VEL3, homolog of VEL1, does similarly regulate gene expression as VEL1, also similar in seed dormancy as in vernalization. Similar works about VRN5, VAL1 etc containing PHD domain indicated a role for the transcriptional repressor in the silencing mechanism (Julia I. Qüesta, et al.,2016, Science).

2) The observation that VEL3 associates with histone deacetylase complexes is interesting, however, more solid evidences are required. VEL3 was able to interact with MSI1, HDAC, but the evidence supporting they work in the same complex is limited. Does VEL3-HDAC and VEL3-PRC2 interaction regulate same gene? Different gene? Or same gene at different stage? VEL3 regulates the deposition of HDAC? Or not. anyway, this part needs more informative results.

3) One of interesting points is VEL3-GFP was primarily located in the nucleolus but a lower GFP signal was observed in the nucleoplasm and VEL3 directed MSI1 to the nucleolus. But the presented data do not provide mechanistic insights about their interaction and function.

4) The authors claimed that this work establishes a transgenerationally inherited maternal epigenetic state controlling progeny seed dormancy. VEL3 is indeed expressed in central cell, which is also associated with seed development (vel3 showed aborted seeds). But the evidence about transgenerational inherit is lack.

Reviewer #2 (Remarks to the Author):

In this manuscript, Chen and colleagues report on the identification of VIN3-LIKE3 (VEL3) as a regulator of endosperm-mediated seed dormancy. They found that VEL3 interacts with the PRC2 subunit MSI1 and showed that VEL3 is required for H3K27me3 establishment at certainly loci in the endosperm. Loss of maternal VEL3 function reduces seed dormancy, while loss of maternal and paternal function causes seed abortion. VEL3 is specifically expressed in the central cell and the endosperm and the phenotypic defects of vel3 mutants a consequence of VEL3 function in either the central cell or the endosperm, or both. Strikingly, the interaction of VEL3 with MSI1 takes place specifically in the nucleolus; however, the connection between this specific localization and gene regulation remains unclear. The authors furthermore identified the histone deacetylase HD2C as interaction partner of VEL3 and found that hd2c mutants have lower dormancy, similar to vel3, but the genetic interaction remains to be clarified. Last, the authors identified ORE1 as a potential downstream gene of VEL3. This paper reports interesting and relevant findings that advance our understanding of seed dormancy control and H3K27me3 establishment in the endosperm. There are however several points that require clarification.

Main points:

1. The title does not reflect what is actually shown in this manuscript. I do not see any evidence for a "transgenerational inherited epigenetic state". The epigenetic modifications requiring VEL3 action are established either in the central cell or the endosperm and are required to establish dormancy in the embryo. This status is not inherited to the next generation, at least this was not shown, so the same generation is affected that is exposed to VEL3 action. There is, therefore, no transgenerational effect and this term should be deleted from the manuscript.

2. Line 18: this part is misleading; whether the established expression patterns are established in the central cell was not shown and "gene expression via PRC2 and HDAC at pericentromeric regions" is an unclear statement. To test whether VEL3 acts in the central cell, seeds from heterozygous vel3 mutants pollinated with wild-type pollen should be tested for their effect on

seed dormancy. It would furthermore need to be tested whether deregulated genes are close to pericentromeric regions, otherwise this statement should be removed.

3. Figure 2, L106: The seed abortion phenotype of *vel3* is interesting, but it remained unclear why the seeds abort. Since the authors imply a functional connection of VEL3 with the PRC2, it would be relevant to test whether the seed abortion is a consequence of endosperm cellularization failure, similar to *fis-prc2* mutants.

4. Based on Figure 4A: Loss of VEL3 has only a specific effect on selected genes marked with H3K27me3. It would be important to know whether those genes are marked by H3K27me3 alone or by H3K27me3/H3K9me2. Since the authors postulate a role of REF6 in regulating genes affected by VEL3, it would be relevant to know whether they can in principle be targeted by REF6. Along those lines it would be relevant to check whether VEL3 targets are downregulated in *ref6*. Also the double mutant phenotype of *vel3ref6* would be informative; if REF6 acts on VEL3 targets, the seed dormancy phenotype of the double mutant should be similar to *vel3*.

5. It should be tested which genes marked by VEL3 become upregulated in *vel3* mutants.

6. Fig 5F: The effect of 35S ORE1 on seed dormancy raises questions. 35S is not active in the endosperm; therefore, the observed effect is unlikely a consequence of increased ORE1 expression in the endosperm.

7. Figure 5G: The *ore1* mutant should be included in this figure to test whether the observed effects are indeed epistatic or rather additive.

8. To test whether HD2C acts in the central cell or endosperm, it would be required to test the effect of heterozygous *hd2c* mutants on seed dormancy (see above comment for *vel3*).

9. VEL3-ChIP data for ORE1, WRKY75, and NAC046 should be included in Fig 5D.

10. (Fig 6C) Do regions with increased H3K9acetylation in *vel3* mutant overlap with regions with decreased H3K27me3? This should be tested.

11. Do *hd2c-1* and *hda19-3* affect H3K9 acetylation on ORE1, WRKY75, and NAC046?

12. Minor points:

13. Fig 4A and Line 124: Based on the methods section, the endosperm H3K27me3 data from early endosperm are from Moreno-Romero et al., rather than Zheng and Gehring 2019 as indicated based on citation. The enrichment of H3K27me3 in pericentromeric regions was also not detected by Zheng and Gehring 2019.

14. L115: The interaction of VEL3 and MSI1 was also found by IP-MS; this data should be mentioned at this position.

15. Figure 4B: it is unclear what the two different colors in figure 4B refer to.

16. Line 26: It should be "In most angiosperms...", since there are angiosperms forming a diploid endosperm

17. Line 29: "or genomic imprinting" is misleading; genomic imprinting leads to parent-of-origin specific expression.

18. L82 and Figure S4: The "structure" should be specified. Based on Figure S4A it is unclear why the TAIR10 annotated cDNA is smaller than the observed VEL3 cDNA, this requires clarification. Furthermore, in the text the authors use the term gVEL3, which is not used in the figure.

19. L.146: the term female germline is misleading; "central cell of the female gametophyte" should be used instead.

20. L153: It is unclear under which conditions these 124 genes become deregulated; after 1 hr or 24 hrs or both?

21. Fig 3A and Fig S3E. The authors claim that VEL3 interacts only with MSI1; however, based on the Y2H data VEL3 and MEA may interact as well. It would be more appropriate to state that "also MEA could interact with VEL3, but the interaction was not as strong as with MSI1."

Reviewer #1 (Remarks to the Author):

This manuscript reported that Arabidopsis VERNALIZATION5/VIN3-LIKE 3 (VEL3) maintains maternal control over progeny seed dormancy by establishing an epigenetic state of histone methylation and acetylation early in endosperm development. VEL3 associates with PRC2 and HDAC complex to establish a trans-generationally inherited repressive epigenetic state. The experiments were well designed and results are interesting. However, I am concerning the novelty of this manuscript.

1) It is not surprised that VEL3 is required for the establishment of H3K27me3 deposition did not give me surprise, which did not highlight the novelty of this work. Five-member VEL gene family was observed in the Arabidopsis genome and VEL1 has been reported acting as component in PHD-PRC2 complex to trigger the epigenetic silencing of FLC during vernalization (Filomena De Lucia et al., 2008, PNAS). VEL3, homolog of VEL1, does similarly regulate gene expression as VEL1, also similar in seed dormancy as in vernalization. Similar works about VRN5, VAL1 etc containing PHD domain indicated a role for the transcriptional repressor in the silencing mechanism (Julia I. Qüesta, et al., 2016, Science).

Response: We agree that VRN5, VEL1 and VIN3 have been understood to form the PHD-PRC2 complex in vernalization response. However VEL3 lacks PHD domain (Mylne *et al.*, Cold Spring Harb Symp Quant Biol 2004. 69: 457-464), so its role for the transcriptional repression may be different from other VEL proteins. H3K27me3 in the endosperm is characterized by its distribution at pericentromeric regions. One key feature of VEL3 is also that it binds to pericentromeric regions, regulates H3K27me3 and H3K9ac at centromere/pericentromeric regions (see revised Figure 7). So we think that VEL3 shows unique function compared to other VEL proteins.

VEL3 is also characterized by its exclusive expression in gametes and early endosperm, establishing primed epigenetic mark in central cell to maternally control seed dormancy. As far as we aware, this is also the first time we reveal the potential role of program cell death-related genes in seed dormancy control through endosperm epigenetics. These indeed advance our understanding in seed dormancy control and endosperm development.

2) The observation that VEL3 associates with histone deacetylase complexes is interesting, however, more solid evidences are required. VEL3 was able to interact with MSI, HDAC, but the evidence supporting they work in the same complex is limited. Does VEL3-HDAC and VEL3-PRC2 interaction regulate same gene? Different gene? Or same gene at different stage? VEL3 regulates the deposition of HDAC? Or not. anyway, this part needs more informative results.

Response: This is indeed an interesting question. To address this we performed H3K9ac ChIP-seq in Col0 and vel3 mutant endosperm. The new data is now included in Figure 7. We show that VEL3 is indeed required for the deacetylation, especially at centromere/pericentromeric regions. Unlike the global reduction of H3K27me3, in the vel3 mutant H3K9ac is re-distributed across the genome, with an unknown mechanism. A similar re-distribution of H3K27me has been previously reported in yeast, which coincides with

domains of heterochromatin marked by H3K9me (Dumesic et al., 2015 Cell). Our data shows that in the absence of VEL3 HDAC complexes lose their peri-centromeric targeting but likely are still catalytically active. We also show that hypoH3K27me3 regions are significantly overlapped with hyperacetylation regions suggesting that the function of VEL3-HDAC is spatially overlapped with the control of H3K27me3.

3) *One of interesting points is VEL3-GFP was primarily located in the nucleolus but a lower GFP signal was observed in the nucleoplasm and VEL3 directed MSI1 to the nucleolus. But the presented data do not provide mechanistic insights about their interaction and function.*

Response: We don't agree with this because it is not previously understood how MSI1 is targeted to different compartments for its various functions. MSI1 has been suggested to associate with PRC2, HDAC and CAF1 complex. Only a subset of MSI1 functions, e.g. deacetylation, occur in the phase separated nucleolus.

4) *The authors claimed that this work establishes a transgenerationally inherited maternal epigenetic state controlling progeny seed dormancy. VEL3 is indeed expressed in central cell, which is also associated with seed development (vel3 showed aborted seeds). But the evidence about transgenerational inherit is lack.*

Response: to address this we removed 'transgenerationally inherited' from the title.

Reviewer #2 (Remarks to the Author):

Main points:

1. *The title does not reflect what is actually shown in this manuscript. I do not see any evidence for a "transgenerational inherited epigenetic state". The epigenetic modifications requiring VEL3 action are established either in the central cell or the endosperm and are required to establish dormancy in the embryo. This status is not inherited to the next generation, at least this was not shown, so the same generation is affected that is exposed to VEL3 action. There is, therefore, no transgenerational effect and this term should be deleted from the manuscript.*

Response: to address this we removed 'transgenerationally inherited' from the title.

2. *Line 18: this part is misleading; whether the established expression patterns are established in the central cell was not shown and "gene expression via PRC2 and HDAC at pericentromeric regions" is an unclear statement. To test whether VEL3 acts in the central cell, seeds from heterozygous vel3 mutants pollinated with wild-type pollen should be tested for their effect on seed dormancy. It would furthermore need to be tested whether deregulated genes are close to pericentromeric regions, otherwise this statement should be removed.*

Response: a good point. To address this we generated heterozygous vel3-1/+ plants and pollinated with Col-0 father. Consistent with the activity of VEL3 in the central cell, the resulting seeds showed an intermediate phenotype between Col-0 and vel3-1. This is not expected if this is a maternal sporophytic inheritance. So we conclude that VEL3 activity in the gametophyte is important. This data is added to Figure 2D. We also deleted the

statement 'gene expression via PRC2 and HDAC at pericentromeric regions' from the manuscript.

3. *Figure 2, L106: The seed abortion phenotype of vel3 is interesting, but it remained unclear why the seeds abort. Since the authors imply a functional connection of VEL3 with the PRC2, it would be relevant to test whether the seed abortion is a consequence of endosperm cellularization failure, similar to fis-prc2 mutants.*

Response: To address this we imaged Col-0 and vel3-1 mutants with Feulgen staining and confocal microscopy. At heart stage we indeed could observe endosperm cellularization in Col0, but not in vel3-1 mutants. This data is added to Figure S3E. So the phenotype resembles fis-prc2 mutants.

4. *Based on Figure 4A: Loss of VEL3 has only a specific effect on selected genes marked with H3K27me3. It would be important to know whether those genes are marked by H3K27me3 alone or by H3K27me3/H3K9me2. Since the authors postulate a role of REF6 in regulating genes affected by VEL3, it would be relevant to know whether they can in principle be targeted by REF6. Along those lines it would be relevant to check whether VEL3 targets are downregulated in ref6. Also the double mutant phenotype of vel3ref6 would be informative; if REF6 acts on VEL3 targets, the seed dormancy phenotype of the double mutant should be similar to vel3.*

Response: To address this we performed additional analysis of our data presented in Figure S5. It seems that VEL3 has a similar effect on H3K27me3 whether or not H3K9me2/3 is also present. We found that a subset can be targeted by REF6. This is now presented in Figure S5. In the main text we draw the conclusion that VEL3 is required for PRC2 function regardless of H3K9me2 status. We also add a venn diagram in Fig.S6 to compare the misregulated genes between vel3-1 and ref6-1 mutants. We concluded that REF6 also regulates some target genes of VEL3, including ORE1. We agree it is interesting to observe the seed dormancy phenotype of vel3ref6 double mutants, but we haven't been able to complete it in the time available.

5. *It should be tested which genes marked by VEL3 become upregulated in vel3 mutants.*

Response: We agree this is useful and the information is now included in supplementary table S3 excel sheet 'VEL3 targets up'.

6. *Fig 5F: The effect of 35S ORE1 on seed dormancy raises questions. 35S is not active in the endosperm; therefore, the observed effect is unlikely a consequence of increased ORE1 expression in the endosperm.*

Response: to address this we set 35S:ORE1 seeds at low temperature to see if they really do have a strong reduced dormancy phenotype as observed in vel3. We found that when set at 16°C 35S:ORE1 does not show a reduced dormancy phenotype. This is consistent with the reviewer's assertion that 35S is not strongly active in the endosperm. The data is now included in Figure 5G.

7. *Figure 5G: The ore1 mutant should be included in this figure to test whether the observed effects are indeed epistatic or rather additive.*

Response: it is now included in Figure 5G.

8. To test whether HD2C acts in the central cell or endosperm, it would be required to test the effect of heterozygous *hd2c* mutants on seed dormancy (see above comment for *vel3*).

Response: we agree this is interesting to do but due to various issues we haven't been able to complete it in the time available.

9. VEL3-ChIP data for ORE1, WRKY75, and NAC046 should be included in Fig 5D.

Response: this data is added to Figure 5D.

10. (Fig 6C) Do regions with increased H3K9acetylation in *vel3* mutant overlap with regions with decreased H3K27me3? This should be tested.

Response: Figure 5D shows that in *vel3* mutant, H3K27me3 is decreased at these genomic regions of ORE1, WRKY75 and NAC046.

11. Do *hd2c-1* and *hda19-3* affect H3K9 acetylation on ORE1, WRKY75, and NAC046?

Response: There was so much to do we have not prioritised these experiments but they are of course relevant to the composition of HDAC complexes affecting expression of those genes.

12. Minor points:

13. Fig 4A and Line 124: Based on the methods section, the endosperm H3K27me3 data from early endosperm are from Moreno-Romero et al., rather than Zheng and Gehring 2019 as indicated based on citation. The enrichment of H3K27me3 in pericentromeric regions was also not detected by Zheng and Gehring 2019.

Response: Actually the data were from Zheng and Gehring but we have here reanalysed our data with that of Moreno-Romero 2019 and included this in figure 4 in this version. It seems for the purposes of VEL3 activity we get a basically similar result from both datasets.

14. L115: The interaction of VEL3 and MSI1 was also found by IP-MS; this data should be mentioned at this position.

Response: We have added a line to this effect in the text.

15. Figure 4B: it is unclear what the two different colors in figure 4B refer to.

Response: Thanks, the legend has been modified to add this information.

16. Line 26: It should be "In most angiosperms...", since there are angiosperms forming a diploid endosperm

Response: Change made.

17. Line 29: "or genomic imprinting" is misleading; genomic imprinting leads to parent-of-origin specific expression.

Response: Corrected.

18. L82 and Figure S4: The “structure” should be specified. Based on Figure S4A it is unclear why the TAIR10 annotated cDNA is smaller than the observed VEL3 cDNA, this requires clarification. Furthermore, in the text the authors use the term gVEL3, which is not used in the figure.

Response: this has been clarified in the legend of Fig S4. Most importantly, exon 2 is shorter in the complementing cDNA and the 3’ end is different. Genbank accession numbers are given so this should be clear.

19. L.146: the term female germline is misleading; “central cell of the female gametophyte” should be used instead.

Response: agreed, and change made.

20. L153: It is unclear under which conditions these 124 genes become deregulated; after 1 hr or 24 hrs or both?

Response: we have clarified this is the endosperm 1hr after imbibition.

21. Fig 3A and Fig S3E. The authors claim that VEL3 interacts only with MSI1; however, based on the Y2H data VEL3 and MEA may interact as well. It would be more appropriate to state that “also MEA could interact with VEL3, but the interaction was not as strong as with MSI1.”

Response: we also saw some autoactivation using MEA so we are not confident in this conclusion.

Reviewer #1 (Remarks to the Author):

The revised manuscript responded well my comments. However, I still have one minor concern about the conclusion of "VEL3 directs MSI1 to the nucleolus", which should be tune down. It is well known that MSI1 was detected in Nucleoplasm and Cytosol. If VEL3 directs MSI1 to the nucleolus, MSI1 should be present in nucleolus in vivo. In addition, the current evidence is still limited for this conclusion.

Reviewer #2 (Remarks to the Author):

The authors made efforts to address my concerns and greatly improved the manuscript. I have one remaining concern, which should be possible to be addressed with the available data by changing the way of presenting the data.

Line 87: ...we pollinated heterozygous /+ plants with wild type seeds"; should be pollen
Fig 2D and lines 88ff: I do not agree that the expected phenotype from crossing *vel3/+* x *wt* is an intermediate phenotype. Instead, if the phenotype is caused by VEL3 acting in the female gametophyte, two seed populations are expected; one, which behaves like wildtype and another, which behaves like homozygous *vel3*. It would be good to test if this prediction holds.

REVIEWERS' COMMENTS

Reviewer #1 (Remarks to the Author):

The revised manuscript responded well my comments. However, I still have one minor concern about the conclusion of "VEL3 directs MSI1 to the nucleolus", which should be tune down. It is well known that MSI1 was detected in Nucleoplasm and Cytosol. If VEL3 directs MSI1 to the nucleolus, MSI1 should be present in nucleolus in vivo. In addition, the current evidence is still limited for this conclusion.

Response: Thanks for pointing this out. In the revised version we have changed in the abstract and main text to 'VEL3 colocalizes with MSI1 in the nucleolus'.

Reviewer #2 (Remarks to the Author):

The authors made efforts to address my concerns and greatly improved the manuscript. I have one remaining concern, which should be possible to be addressed with the available data by changing the way of presenting the data.

Line 87: ...we pollinated heterozygous +/- plants with wild type seeds"; should be pollen

Response: This has been changed in the main text.

Fig 2D and lines 88ff: I do not agree that the expected phenotype from crossing vel3/+ x wt is an intermediate phenotype. Instead, if the phenotype is caused by VEL3 acting in the female gametophyte, two seed populations are expected; one, which behaves like wildtype and another, which behaves like homozygous vel3. It would be good to test if this prediction holds.

Response: We think the reviewer is mistaken. If you mix two populations of seeds with different germination frequencies in equal proportions then the obvious result is a mixed population whose mean germination rate is precisely intermediate between the means of the two starting populations.

Other changes requested by the editor have been actioned, please see revised manuscript with changes tracked for details.